# Randomized Clinical Trials and Observational Tribulations: Providing Clinical Evidence for Personalized Surgical Pain Management Care Models

**DOI:** 10.3390/jpm13071044

**Published:** 2023-06-25

**Authors:** Ivo Abraham, Kai-Uwe Lewandrowski, John C. Elfar, Zong-Ming Li, Rossano Kepler Alvim Fiorelli, Mauricio G. Pereira, Morgan P. Lorio, Benedikt W. Burkhardt, Joachim M. Oertel, Peter A. Winkler, Huilin Yang, Jorge Felipe Ramírez León, Albert E. Telfeian, Álvaro Dowling, Roth A. A. Vargas, Ricardo Ramina, Marjan Asefi, Paulo Sérgio Teixeira de Carvalho, Helton Defino, Jaime Moyano, Nicola Montemurro, Anthony Yeung, Pietro Novellino

**Affiliations:** 1Pharmacy Medicine, and Clinical Translational Sciences, University of Arizona, Roy P. Drachman Hall, Rm. B306H, Tucson, AZ 85721, USA; abraham@pharmacy.arizona.edu; 2Center for Advanced Spine Care of Southern Arizona, Tucson, AZ 85712, USA; 3Department of Orthopaedics, Fundación Universitaria Sanitas, Bogotá 111321, Colombia; 4Department of Orthopedics, Hospital Universitário Gaffre e Guinle, Universidade Federal do Estado do Rio de Janeiro, Rio de Janeiro 20270-004, Brazil; 5Department of Orthopaedic Surgery, College of Medicine—Tucson Campus, Health Sciences Innovation Building (HSIB), University of Arizona, 1501 N. Campbell Avenue, Tower 4, 8th Floor, Suite 8401, Tucson, AZ 85721, USA; elfar@arizona.edu (J.C.E.); lizongming@arizona.edu (Z.-M.L.); 6Department of General and Specialized Surgery, Gaffrée e Guinle University Hospital, Federal University of the State of Rio de Janeiro (UNIRIO), Rio de Janeiro 20270-004, Brazil; fiorellirossano@hotmail.com; 7Faculty of Medicine, University of Brasilia, Federal District, Brasilia 70919-900, Brazil; mauriciogpereira@gmail.com; 8Advanced Orthopaedics, 499 E. Central Pkwy, Ste. 130, Altamonte Springs, FL 32701, USA; mloriomd@gmail.com; 9Wirbelsäulenzentrum/Spine Center—WSC, Hirslanden Klinik Zurich, Witellikerstrasse 40, 8032 Zurich, Switzerland; benedikt.burkhardt@gmail.com; 10Klinik für Neurochirurgie, Universität des Saarlandes, Kirrberger Straße 100, 66421 Homburg, Germany; Joachim.Oertel@uks.eu; 11Department of Neurosurgery, Charite Universitaetsmedizin Berlin, 13353 Berlin, Germany; prof.peter.winkler@gmail.com; 12Orthopaedic Department, The First Affiliated Hospital of Soochow University, No. 899 Pinghai Road, Suzhou 215031, China; suzhouspine@163.com; 13Minimally Invasive Spine Center Bogotá D.C. Colombia, Reina Sofía Clinic Bogotá D.C. Colombia, Department of Orthopaedics, Fundación Universitaria Sanitas, Bogotá 110141, Colombia; jframirezl@yahoo.com; 14Department of Neurosurgery, Rhode Island Hospital, The Warren Alpert Medical School of Brown University, Providence, RI 02903, USA; ATelfeian@Lifespan.org; 15Department of Orthopaedic Surgery, University of São Paulo, Ribeirão Preto 14071-550, Brazil; adowling@dws.cl (Á.D.); hladefin@fmrp.usp.br (H.D.); 16Department of Neurosurgery, Foundation Hospital Centro Médico Campinas, Campinas 13083-210, Brazil; rothvargas@hotmail.com; 17Neurological Institute of Curitiba, Curitiba 80230-030, Brazil; ramina@hospitalinc.com.br; 18Department of Biology, Nano-Biology, University of North Carolina, Greensboro, NC 27413, USA; massefi@aggies.ncat.edu; 19Pain and Spine Minimally Invasive Surgery Service, Gaffre e Guinle University Hospital, Rio de Janeiro 20270-004, Brazil; paulo.carvalho@unirio.br; 20La Sociedad Iberolatinoamericana De Columna (SILACO), The Spine Committee of the Ecuadorian Society of Orthopaedics and Traumatology (Comité de Columna de la Sociedad Ecuatoriana de Ortopedia y Traumatología), Quito 170521, Ecuador; jaime.moyano7@icloud.com; 21Department of Neurosurgery, Azienda Ospedaliero Universitaria Pisana, University of Pisa, 56124 Pisa, Italy; nicola.montemurro@unipi.it; 22Desert Institute for Spine Care, Phoenix, AZ 85020, USA; ayeung@sciatica.com; 23Guinle and State Institute of Diabetes and Endocrinology, Rio de Janeiro 20270-004, Brazil; pietro.novellino@hotmail.com

**Keywords:** surgical clinical trials, personalized care models, pain generators, clinical evidence

## Abstract

Proving clinical superiority of personalized care models in interventional and surgical pain management is challenging. The apparent difficulties may arise from the inability to standardize complex surgical procedures that often involve multiple steps. Ensuring the surgery is performed the same way every time is nearly impossible. Confounding factors, such as the variability of the patient population and selection bias regarding comorbidities and anatomical variations are also difficult to control for. Small sample sizes in study groups comparing iterations of a surgical protocol may amplify bias. It is essentially impossible to conceal the surgical treatment from the surgeon and the operating team. Restrictive inclusion and exclusion criteria may distort the study population to no longer reflect patients seen in daily practice. Hindsight bias is introduced by the inability to effectively blind patient group allocation, which affects clinical result interpretation, particularly if the outcome is already known to the investigators when the outcome analysis is performed (often a long time after the intervention). Randomization is equally problematic, as many patients want to avoid being randomly assigned to a study group, particularly if they perceive their surgeon to be unsure of which treatment will likely render the best clinical outcome for them. Ethical concerns may also exist if the study involves additional and unnecessary risks. Lastly, surgical trials are costly, especially if the tested interventions are complex and require long-term follow-up to assess their benefit. Traditional clinical testing of personalized surgical pain management treatments may be more challenging because individualized solutions tailored to each patient’s pain generator can vary extensively. However, high-grade evidence is needed to prompt a protocol change and break with traditional image-based criteria for treatment. In this article, the authors review issues in surgical trials and offer practical solutions.

## 1. Introduction

The traditional clinical trial design calls for standardization, objective outcome measures, adequate sample size, randomization, blinding, long-term follow-up, statistical analysis, and collaboration with data sharing to improve a trial’s quality [1]. These requirements are rooted in drug-trial concepts. Standardizing surgical [2] and interventional pain management techniques [3] aims to reduce the variability between surgeons and ensures that procedures are performed consistently for the same clinical indications. This standardization is essential, as it helps to reduce bias by having better control of confounding variables. However, whether randomized trials should be the exclusive design of choice in surgical trials is not evident and can be challenged on different fronts: design, surgical-center selection, sampling, standardization of treatment with or without personalization to the patient, treatment implementation under various conditions of surgical experience and skill, measurement of objective and patient-centric self-reported outcomes, duration of follow-up to capture both short- and long-term outcomes and adverse events, and data management and statistical analyses including confounder management and data sharing [2,4,5,6,7,8,9,10]. In this paper, we review several issues related to the design, implementation, and analysis of controlled trials and non- (or minimally) controlled observational studies, using personalized pain management as the exemplar.

## 2. How to Manage Skill in Surgical Trials

Surgeons’ skill levels can vary widely. Managing this in a surgical trial involves assessing each surgeon’s ability to operate consistently at a high quality. Several steps can be taken to address surgical skill variations (Table 1). Surgical trial procedures should be standardized and follow a written protocol to ensure that all surgeons perform the same procedure. Pre-trial training is one way to familiarize surgeons with the surgical technique and procedural steps. Trial surgeons should be skilled and selected based on their qualifications, experience, and previous surgical outcomes. The trial outcomes should be monitored regularly to identify variations in surgical skill level, and participating surgeons should receive feedback to improve their technique when needed [11].

Randomization of surgeons can also be considered, but may be impractical unless the trials are carried out in large departments with multiple qualified surgeons or the patients are enrolled at several institutions in multi-center trials. At a minimum, surgeons should be evenly assigned across the study groups. In the rare cases possible, surgeons should also be blinded to the treatment allocation to reduce bias due to differences in surgical skills between the surgeons [11]. Many surgical trials may suffer from being unable to fulfill all these requirements. In personalized surgical pain management care models, where the surgeon has identified the pain generator using peer-reviewed published protocols, many measures to reduce surgeon-induced bias may be impractical.

Managing surgical skills in surgery trials is critical for ensuring trial outcomes’ validity and dependability. Surgeon training and standardization, surgeon selection, centralization of procedures, monitoring of surgical performance, and assessment of surgical skills are all noticeable strategies for managing surgical skills in surgical trials [12]. Implementing these techniques will likely enhance surgical skill management and reduce variability in surgical complications, enhancing the accuracy and reliability of surgical trials [12]. Additionally, clinical trial errors during surgery can affect the validity and reliability of trial data, possibly leading to inaccurate findings regarding the safety and efficacy of surgical interventions. Selection bias, lack of blinding, variability in surgical technique, difficulty in standardizing outcome measures, limited sample size, inadequate study design and methodology, deficient statistical analysis, and lack of long-term follow-up are some of the main errors in surgical clinical trials [13].

## 3. Major Errors in Clinical Trials in Surgery

The quality of clinical trials in surgery may suffer from several significant errors (Table 1). Results may be biased and skewed if inappropriate candidates for the surgical procedure are enrolled. Therefore, inclusion and exclusion criteria should be well thought-out—yet too-stringent criteria may distort the study population to a point where it no longer represents the general population. Inconsistencies in surgical technique may be another source of error, making data interpretation difficult. Insufficient follow-up and the lack of blinding may lead to a biased outcome assessment due to the placebo effect or other factors. Incomplete reporting of missing information is widespread in surgical trials, since the outcome is often unknown until several years after the index procedure [2]. Many surgical patients are lost in follow-up. Attrition of the surgical study population in follow-up may also reduce the statistical power if the numbers of enrolled patients with complete follow-up are too low. Therefore, it may be difficult to detect meaningful differences (Figure 1).

Most studies in surgical research are retrospective longitudinal observational cohort studies without a control group. Interpretation of the study results may be difficult without a control group or with a control group that is not well-matched to the treatment group.

## 4. Differences between Drug Trials and Surgical Trials

There are several critical differences between drug and surgical clinical trials. The type of intervention studied is entirely different. In drug trials, a medication or drug therapy is investigated. In surgical trials, the intervention is a surgical procedure. A placebo control group is often used to compare the effects of the drug with those of a placebo [14]. A sham surgical procedure may be used as a control group in surgical trials. However, this is rarely approved by institutional review boards (IRB) because of ethics concerns. Blinding study personnel and patients in a drug trial is typically not problematic. However, it is nearly impossible in surgical trials. The primary outcome in drug trials is often based on laboratory or clinical tests, while in surgical trials, the outcomes are typically based on clinical observations and patient-reported outcomes [10]. Therefore, surgical trials are often much longer than drug trials. Their sample sizes are often limited by access to patients, resources, and cost. The threshold for patients to enroll in drug trials is lower because of fewer immediate risks. In surgical trials, patients are exposed to more significant risks because of bleeding, infection, and other peri- or post-operative complications.

## 5. Why Are Patients Reluctant to Participate in Surgical Trials?

Patients may be reluctant to participate in surgical trials primarily because of the fear of the unknown, since surgical trials may involve new or experimental procedures; perceived higher risk of complications and adverse effects of new surgical procedures; fear of significant disruption to a patient’s daily life with untested surgical procedures; or hesitancy to participate in a trial that may require extended hospital stays or recovery periods [15,16]. Being confronted with the complexity of the surgical trial may also drive away patients, particularly those with a limited understanding of the study design, procedures, or potential benefits and risks associated with the trial. Others may not trust the medical establishment due to previous negative experiences with healthcare providers (Table 2).

Patients may also consider confidentiality and privacy concerns significant deterrents to trial participation since it requires sharing personal information or medical history. Patients may be concerned about incurring additional costs, including co-pays, deductibles, and other out-of-pocket expenses due to trial participation. Trial administrators may improve patient enrollment by addressing these hurdles. It is important for researchers to address these concerns and communicate effectively with potential study participants to help them make informed decisions about trial participation [17]. This may involve providing detailed information about the study design and procedures, addressing concerns about risks and benefits, and ensuring confidentiality and privacy protections.

## 6. Why Do Surgeons Not Like to Do Clinical Trials?

Surgeons may be hesitant to participate in clinical trials for various reasons, including concerns about patient safety, the complexity and time commitment involved in a trial, and the potential impact on their clinical practice [15]. Surgeons also want to avoid harming their patients, which may not be clearly guaranteed upfront, particularly when considering experimental treatments. Using unfamiliar techniques or equipment can be challenging and potentially lead to complications. Adverse events may also drive referrals from their practice (Table 3). 

Clinical trials may also force surgeons to deviate from their usual practice, which may provoke errors. Clinical trials are also a significant time commitment and essentially an uncompensated task which, in today’s healthcare environment, could be economically detrimental to their practice. Additional costs may arise from the required rigorous planning, execution, and follow-up, which can be challenging to balance with a busy surgical practice unless reimbursed by the trial sponsors (if there are any). Clinical trials may also distract from surgeons’ practice and limit their ability to see patients, which could amount to additional revenue losses, due to lost opportunity costs and futures. 

## 7. Should We Even Perform Randomized Clinical Trials in Surgery?

Randomized trials can be a valuable tool for evaluating the effectiveness of surgical treatments and for identifying potential risk of complications arising from the new treatment being tested, thus providing valuable insights into the safety and efficacy of surgical interventions. However, maintaining a controlled environment to minimize bias can be challenging (Table 4). Notably, bias in selecting participants; blinding them as well as surgeons, other clinicians, and researchers to group assignment and the intervention being studied; and the ethical considerations of exposing patients to potential risks associated with surgical procedures are all challenges in surgical RCTs.

Other factors making RCTs in surgery difficult include variability in surgical techniques as well as lack of standardized outcome measures, whether objectively assessed outcomes, clinician appraisals, or patient-reported outcomes [18]. Table 4 also lists various strategies to enhance clinical trials in surgery, including standardization of surgical techniques, use of patient-reported outcomes, blinding, use of objective outcome measures, use of multi-center trials, and collaboration between surgeons and researchers. 

## 8. Opportunities and Limitations of Observational Studies Compared to Clinical Trials

Observational studies may provide valuable insights into real-world surgical practice and its outcomes. Such studies evaluate the real-world effectiveness of surgical methods outside the controlled environment of an RCT, when applied across patient populations that vary in comorbidities, severity of disease, and physical factors that may complicate the planned surgery. Thus, observational studies bridge the gap between controlled clinical trials and the diffusion of a novel surgical approach into everyday surgical practice. Further, new surgical methods often require long-term follow-up to fully assess their efficacy and safety. By tracking patients over extended periods of time, observational studies generate data on both short- and long-term outcomes and complications and thus inform us about the robustness and durability of novel surgical methods. This enables identification of potential risks, benefits, mediating and moderating factors of outcome, as well as unanticipated consequences and outcomes. Observational studies also enable comparative effectiveness research on different surgical methods, using primary clinical data as well as secondary data from electronic medical records and administrative databases. On the front end of surgical innovation, observational studies may serve as pilot studies to inform subsequent randomized trials by evaluating the appropriateness, relevance, feasibility, and safety of novel surgical methods. Observational studies may be the only way to study the effectiveness and safety of surgical methods that target rare diseases or other patient populations with low prevalence. From a practical perspective, observational studies are much easier to orchestrate in a more cost-effective manner.

Conversely, however, observational studies are considered lower-quality studies for investigating the effectiveness and safety profile of clinical interventions including surgeries [19]. Their major drawbacks are the lack of randomization, the inherent equal distribution of potential known and unknown confounders across the study groups, and the introduction of bias by investigators with poor control of confounding factors (Figure 2). 

However, observational studies have some advantages over RCTs. They are less disruptive to the surgeon’s clinical practice, as the new treatment to be studied may be an iteration of established protocols or comparable to familiar surgical protocols. It is also much easier to enroll a large number of patients, as surgeons can draw from a more diverse patient population more reflective of real-world patients. However, building the sample for an observational surgery study needs to be undertaken carefully to avoid other selection biases: surgeon’s preference, experience, and expertise; patient characteristics that may favor or disfavor the surgical method being evaluated; or the effect of unmeasured confounding variables that can impact effectiveness and safety outcomes and are independent of and unrelated to the surgical method being evaluated. Further, there is less control over the surgical method being studied. Problems with skill level, inability to double-blind, and issues with the practicality of randomization create the well-recognized “glass-ceiling” effect, where well-orchestrated and executed observational surgical studies may be graded as higher-grade clinical evidence than poorly executed prospective randomized clinical trials [20,21,22]. Observational studies may be subject to measurement bias, as they often collect outcome data observed and rated by the surgeon, other clinical staff, or research staff. Data collection methods may be less rigorously operationalized, variables may be inadequately defined, follow-up periods may vary in length, or other clinically unrelated censoring events may occur. In principle, causality cannot be inferred, though applying strict conditions and advanced statistical methods may allow for speculation, if not establishment of a causal association. The results of any observational study need to be interpreted with caution, and limitations need to be carefully appraised when interpreting its results.

## 9. Providing Evidence on Surgical Methods for Which There Are No RCTs

Personalized interventional and surgical pain management concepts focused on treating validated pain generators with minimally invasive and endoscopic surgical techniques are recommended when no RCT is possible. There are no randomized clinical trials proving that targeted, individualized care is more cost-effective and delivers better long-term clinical outcomes than a population-based care model where many patients do not receive any treatments until the end-stage of the disease because they do not meet the criteria of medical necessity for treatment. Treating patients in pain who are either too young, too old, or suffer from medical comorbidities are at the heart of the personalized medicine approach. When clinical trials on innovative techniques are unavailable, surgeons should rely on the best available evidence. Many innovations start with personal observations, published case reports, observational studies, and expert opinions. Surgeons should consult key opinion leaders, attend workshops and conferences, and participate in discussions on surgical forums. They should find mentor surgeons who can help gather insights and opinions from other experts, weigh the risks versus the benefits of new innovative techniques, and use caution when there is limited evidence available on a surgical method. It is also essential to fully inform patients of the potential benefits and risks associated with a new procedure and explain the motivation for trialing a new technique. 

## 10. Discussion

It is important to frame the major tenets of this paper within the (potentially conflicting) context of standardized surgical procedures and individual patient needs. Surgical procedures often need adjustment in order to accommodate patient-specific needs, clinical challenges that a patient’s condition may bring to the operating table, or other patient-centric factors that require adaptation to assure the best possible outcome. In an era of population-based health care procedures, it is essential to balance cost-responsible care with individual patient needs.

Another balance to be achieved is between the effectiveness outcomes of the surgery and outcome measures important to patients: survival rates, disease control versus progression, perceived alleviation of disease-related burdens, quality of life, and patient satisfaction. This calls for enabling (and paying for) personalized care in which surgeries may need to go beyond the core operation procedures and add other (or, subtract some and add other) procedures or variants that may add surgical time and effort to achieve patient-centric outcomes. Patient-reported outcome measures (PROMs) can provide valuable insights into patients’ experiences and perceptions [5].

Arguably, the incremental benefits from such personalized surgical care may yield better outcomes that, in turn (and notwithstanding any incremental cost of adding the personalized care), will diminish or eliminate the need for follow-up medical or surgical care and the associated costs. Satisfied patients whose personal treatment goals are achieved in conjunction with their surgeon’s treatment goals are more likely to stop utilizing medical services for the same condition. 

To return to our earlier discussion of controlled and non-controlled studies, observational cohort studies of real-world surgical practice patterns and associated outcomes should be standard practice in building the evidence base for surgical innovation—complementing any evidence from RCTs or, in the absence of RCTs, providing substantive evidence of surgical innovation. This certainly applies to our call for comparing standardized surgical procedures with and without personalized care to assess whether the patient-specific personalization yields superior short-term and long-term objective clinical outcomes (as judged by the surgeon and by the patient). A critical methodological issue concerns the comparability of the “with” and “without” study arms in baseline characteristics. Included in the data study data model. However, the propensity model will be inherently limited to known and explicated confounders that are included; unlike randomization, where it is assumed that also unknown confounders will be balanced [23,24].

In addition, by considering (a plethora of) patients’ genetic, medical, and behavioral characteristics, it will be possible to go beyond estimating average effects for a population at large and instead categorize patients into subpopulations that vary in their symptomatology due to painful conditions or in their response to a therapeutic intervention. In turn, this will make it possible to determine who will benefit from a targeted therapy or intervention the most [25,26,27]—and who most likely will not. 

Such comparative studies should also evaluate patients’ post-surgery utilization of health care services in both the short and long term, as well as the associated budget impact. For instance, does a personalized care approach lead to less health care utilization and thus save costs? Does it lead to different health care utilization patterns, with some services no longer necessary but other services added, yet still remain either cost-saving or budget-neutral? Administrative data, such as those from electronic health records or claims data, may prove helpful in this investigation, though with due reservation about potential inherent biases. Data permitting, formal cost-effectiveness studies should be performed. These should evaluate the cost differentials, decremental or incremental, of standard surgery with versus without personalized care in not only gaining better positive outcomes, but also, in separate analyses, of averting negative outcomes—in both the short and long term.

Lastly, the results of both controlled and observational studies should be evaluated independently at several levels. First, while common for RCTs, an independent Data Safety and Monitoring Board should also be appointed in observational studies to follow the trial from pre-launch through implementation, trial closure, database lock, and statistical analysis. The second level concerns external peer review by highly ranked journals; followed, thirdly, by expert consensus review and, fourthly, by guideline-setting organizations.

In this data-intensive, personalized medicine era, traditional population-based RCTs, which compare average responses between treatment arms, may play a smaller role, and studies comparing personalized versus non-personalized interventions may become commonplace. This paradigm shift will better document the clinical and patient-centric advantages and the economic implications of targeted interventional and surgical pain management therapies aimed at treating validated pain generators [28]. Intervention-matching trials and large-scale, data-driven systems analyses will lead to the more rapid discovery of predictors of favorable clinical outcomes for each patient. This new evidence can then be formalized in differentiated surgical treatment models and algorithms that match individualized variants of surgical procedures to specific patients in a patient-centric and cost-responsible way.

## Figures and Tables

**Figure 1 jpm-13-01044-f001:**
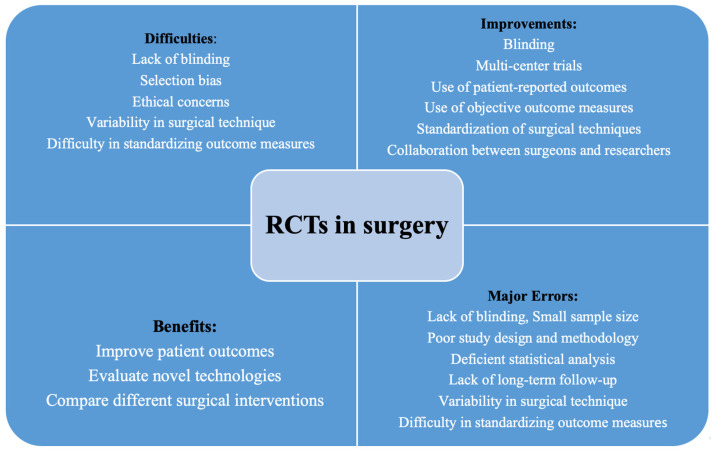
Difficulties, improvements, major errors, and benefits of randomized clinical trials (RCTs) of surgical interventions.

**Figure 2 jpm-13-01044-f002:**
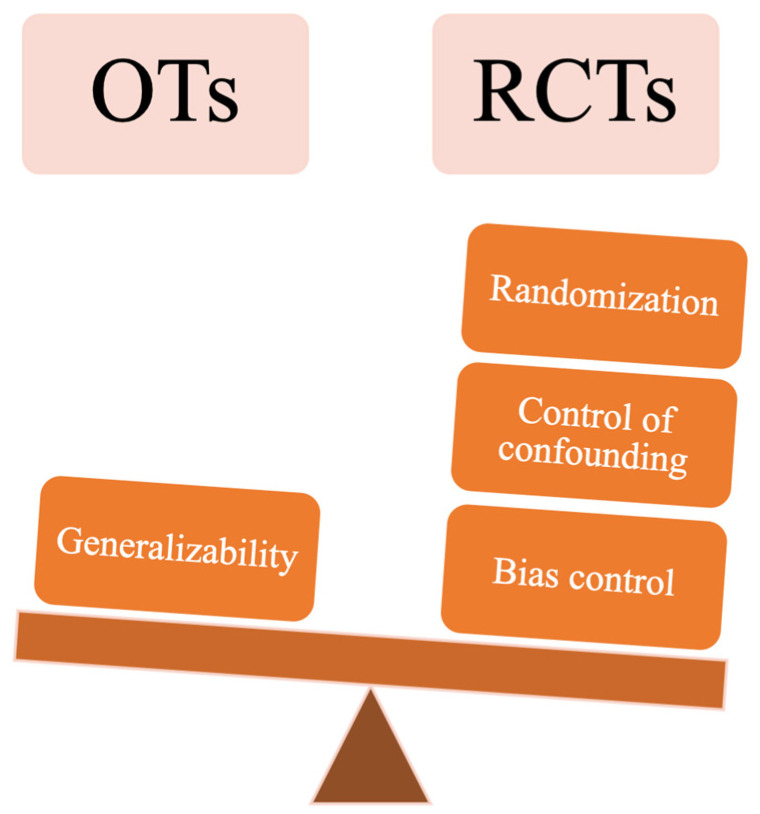
The main differences between observational trials (OTs) in comparison with randomized clinical trials (RCTs).

**Table 1 jpm-13-01044-t001:** Managing skill level variation in surgical RCTs.

	Problem	Description
Managing surgical skills	Surgeon training and standardization	Educating surgeons and standardizing surgical methods can assist in minimizing surgical skill variability and improve the efficacy of surgical trials.
	Surgeon selection	Choosing experienced and highly qualified surgeons to join investigations may reduce variability of surgical results.
	Centralization of procedure	Centralizing treatments to a few competent surgeons can guarantee that surgical interventions are conducted consistently and skillfully.
	Monitoring of surgical performance	Surgical-skill-related problems can be found and resolved by regular monitoring of surgical performance, such as by employing surgical checklists.
	Assessment of surgical skill	Integrating objective measurements of surgical ability, such as time-motion assessment or surgical evaluation tools, can guarantee that surgical operations are continuously and effectively performed.
Errors	Selection bias	A clinical trial’s sample may be biased if patients who are candidates for surgery are not eligible to participate.
	Blinding	In many surgical trials, it can be challenging to blind the patient and surgeon to the intervention, which can result in observer bias.
	Variability in surgical technique	Surgical procedures and approaches can differ substantially amongst surgeons, which can make controlling this source of variation in a clinical trial challenging.
	Difficulty in standardizing outcome measures	Surgical results can be hard to measure and are susceptible to observer bias, making it difficult to develop accurate and valid measurements in surgical trials.
	Small sample size	Several surgical trials have limited sample numbers, which makes detecting substantial differences in outcomes across groups problematic.
	Poor study design	The reliability and accuracy of trial outcomes can be affected by poorly designed studies, such as those with insufficient randomization or an absence of a control group.
	Lack of long-term follow-up	Numerous surgical trials lack proper long-term follow-up, which makes it impossible to evaluate the long-term advantages and disadvantages of a surgical intervention.

**Table 2 jpm-13-01044-t002:** Patient barriers to participating in RCTs.

Barrier	Description
Lack of understanding	Most patients seem to be unaware of the benefits and risks of joining RCTs.
Fear of being a “guinea pig”	There are patients who might be hesitant to participate in RCTs because they believe that they will be given unproven or experimental healthcare.
Concerns about receiving a placebo	Patients might be hesitant to provide informed permission for surgical trials because they are concerned about being assigned to a control group and not receiving the complete surgical treatment.
Logistical issues	Individuals who may find it challenging to engage in RCTs would conclusively require more time and transportation.
Health insurance	Some patients may not have enough health insurance or may be concerned about the costs of participating in RCTs.
Trust in the medical community	Whether they have doubts about the medical profession or fear of being taken advantage of, patients can be hesitant to take part in clinical research.

**Table 3 jpm-13-01044-t003:** Surgeon barriers to participating in clinical trials.

Lack of standardization	In clinical trials, surgical techniques must be standardized in order to account for variability and deliver accurate results. Some surgeons may argue that standardization restricts their ability to perform surgery in the manner in which they believe is best for their patients.
Time constraints	For busy surgeons, clinical studies can be time-consuming and may require extra follow-up and documentation.
Financial considerations	Clinical trials may not be as economically advantageous as ordinary surgical interventions are, and surgeons may be unwilling to take part if they believe that it will have a detrimental influence on their profession or their practice.
Perception of research	Clinical trials may not be viewed by many surgeons as a useful and important part of surgical practice, but rather as a merely academic exercise.
Ethical considerations	It can be challenging to obtain patients’ informed consent for surgical trials, and there can be ethical issues in restraining a patient from a potentially helpful surgical intervention.

**Table 4 jpm-13-01044-t004:** Surgical RCTs Problems and Solutions.

	Factors	Description
Difficulties	Selection bias	Patients who are surgical candidates may not be eligible for a clinical study, resulting in a biased sample.
	Blinding	It is challenging to blind the patient and surgeon to the intervention in many surgical studies, which causes bias on the part of the observers.
	Ethical concerns	Patients’ informed consents may be hard to obtain, and there may be ethical concerns regarding rejecting a patient from potentially helpful surgical intervention.
	Variability in surgical technique	Surgical procedures and strategies can differ substantially amongst surgeons, and this source of variation is difficult to manage in a clinical study.
	Difficulty in standardizing outcome measures	Surgical consequences can be hard to measure and are susceptible to observer bias, making it difficult to develop precise and reliable outcome data in surgical trials.
Improving factors	Standardization of surgical techniques	The development and promotion of standardized surgical procedures and practices can help with decreasing variability and improving the level of surgical trials.
	Use of patient-reported outcomes	A more complete picture of the advantages and disadvantages of a surgical intervention can be obtained by incorporating patient-reported outcomes into surgical trials.
	Blinding	To lessen observer bias, efforts should be taken to obliterate knowledge of the intervention from the patients’ and surgeons’ perspectives.
	Use of objective outcome measures	Using objective data in surgical trials, such as imaging findings or the outcomes of biopsies, can lessen observer bias and improve the accuracy of outcome measurements.
	Multi-center trials	Conducting multi-center tests can improve the outcome generalization and overcome the issue of limited sample numbers that might emerge in single-center trials.
	Collaboration between surgeons and researchers	Collaborations between surgeons and scientists can aid in the design and conduct of high-quality surgical trials.

## Data Availability

The data presented in this study are publicly available.

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
