# Peer review of "Randomized Clinical Trials and Observational Tribulations: Providing Clinical Evidence for Personalized Surgical Pain Management Care Models"

_jpm, 2023, doi:10.3390/jpm13071044_

Round 1
Reviewer 1 Report
The manuscript focuses on an important topic: How should we gain information regarding outcome from surgical procedures. It is the same topic that Bogduk and Fraifels [1] faced in their article 2010, but still equally important.
Many of the aspects discussed are true, but I would like to have more information about observational studies and how to make those more usable and giving them higher value in an evidence based evaluation. Here a discussion about propensity score and the methodology using propensity score weighting in order to adopt an observational study to mimic randomized trials would be appreciated [2,3,4]
If those parts are included I think the manuscript can be of good help for surgeons.
1. Bogduk N, Fraifeld EM. Proof or consequences: who shall pay for the evidence in pain medicine? Pain Med. 2010 Jan;11(1):1–2. 2. Lunceford JK, Davidian M. Stratification and weighting via the propensity score in estimation of causal treatment effects: a comparative study. Stat Med. 2004 Oct 15;23(19):2937–60. 3. Austin PC, Mamdani MM. A comparison of propensity score methods: a case-study estimating the effectiveness of post-AMI statin use. Stat Med. 2006 Jun 30;25(12):2084–106. 4. Lunceford JK. Stratification and weighting via the propensity score in estimation of causal treatment effects: a comparative study. Stat Med. 2017 Jun 30;36(14):2320.Good
Author Response
Reviewer #1:
“The manuscript focuses on an important topic: How should we gain information regarding outcome from surgical procedures. It is the same topic that Bogduk and Fraifels [1] faced in their article 2010, but still equally important.
Many of the aspects discussed are true, but I would like to have more information about observational studies and how to make those more usable and giving them higher value in an evidence based evaluation. Here a discussion about propensity score and the methodology using propensity score weighting in order to adopt an observational study to mimic randomized trials would be appreciated [2,3,4]
If those parts are included I think the manuscript can be of good help for surgeons.
- Bogduk N, Fraifeld EM. Proof or consequences: who shall pay for the evidence in pain medicine? Pain Med. 2010 Jan;11(1):1–2.
- Lunceford JK, Davidian M. Stratification and weighting via the propensity score in estimation of causal treatment effects: a comparative study. Stat Med. 2004 Oct 15;23(19):2937–60.
- Austin PC, Mamdani MM. A comparison of propensity score methods: a case-study estimating the effectiveness of post-AMI statin use. Stat Med. 2006 Jun 30;25(12):2084–106.
- Lunceford JK. Stratification and weighting via the propensity score in estimation of causal treatment effects: a comparative study. Stat Med. 2017 Jun 30;36(14):2320.”
Response:
We edited the manuscript extensively to address this reviewer’s comments and added the 4 suggested references. To better illustrate the extensive revisions we left the track change feature on so the reviewers can easily identify the requested changes. We can reformat the manuscript at a later stage for the publisher. We reformatted the manuscript and added language to address this reviewer’s request for more detailed discussion regarding observational study and propensity scoring to mimic randomized trials. The manuscript is already 16 pages long and the subject matter is complex. We hope that this reviewer finds out edits and additions satisfactory to recommend publication.
Reviewer 2 Report
This is a very interesting paper highlighting the barriers for conducting an RCT in surgery. The first part could serve as an excellent editorial. However, the manuscript's title is "PROVIDING CLINICAL EVIDENCE FOR PERSONALIZED 2 SURGICAL PAIN MANAGEMENT CARE MODELS" and pain management is only mentioned last and briefly. Moreover, the transition from difficulties in conducting RCTs to pain and relevant suggestions is not smooth. Discussion is short and general, could be missed and not diminish the general aspect of the manuscript.
Author Response
Response:
We edited the manuscript extensively to address this reviewer’s comments. To better illustrate the extensive revisions we left the track change feature on so the reviewers can easily identify the requested changes. We reformatted the manuscript and added language to address this reviewer’s request for more pain management discussion and smoother transitions between the segments. Many of the issues between surgical and pain management clinical trials are similar and the others are of the opinion that there is increasing overlap between these two subspecialties with the advent of minimally invasive, endoscopic and interventional pain surgery. A new subspecialty of interventional pain surgery may be emerging.
We changed the title to read: “RANDOMIZED CLINICAL TRIALS AND OBSERVATIONAL TRIBULATIONS : PROVIDING CLINICAL EVIDENCE FOR PERSONALIZED SURGICAL PAIN MANAGEMENT CARE MODELS” and added language regarding observational study and propensity scoring to mimic randomized trials. The manuscript is already 16 pages long and the subject matter is complex. We hope that this reviewer finds out edits and additions satisfactory to recommend publication. However, we would be glad to expand the discussion further if deemed necessary by this reviewer.
Lastly, we appreciate this reviewer’s thorough review of our manuscript.
Round 2
Reviewer 2 Report
Thank you for considering my review. I have no further comments.
Author Response
Thank you very much.